# Phosphorylated tau 181 and 217 are elevated in serum and muscle of patients with amyotrophic lateral sclerosis

Samir Abu-Rumeileh[1,11], Leila Scholle[1,11], Alexander Mensch [1], Henning Großkopf [1], Antonia Ratti [2,3], Anna Kölsch[1], Gisela Stoltenburg-Didinger[1,4], Julian Conrad[5], Anna De Gobbi[2], Lorenzo Barba[1], Petra Steinacker[1], Hans-Wolfgang Klafki [6], Patrick Oeckl [7,8], Steffen Halbgebauer [7,8], Caroline Stapf[1], Andreas Posa[1], Thomas Kendzierski[1], Vincenzo Silani [2,9], Lucrezia Hausner [10], Nicola Ticozzi [2,9], Lutz Froelich [10], Jochen Hans Weishaupt[5,7], Federico Verde[2,9] & Markus Otto [1] ✉

Blood phosphorylated (p)-tau 181 and p-tau 217 have been proposed as accurate biomarkers of Alzheimer's disease (AD) pathology. However, blood p-tau 181 is also elevated in amyotrophic lateral sclerosis (ALS) without a clearly identified source. We measured serum p-tau 181 and p-tau 217 in a multicentre cohort of ALS (n = 152), AD (n = 111) cases and disease controls (n = 99) recruited from four different centres. Further, we investigated the existence of both p-tau species using immunohistochemistry (IHC) and mass spectrometry (MS) in muscle biopsies of ALS cases (IHC: n = 13, MS: n = 5) and disease controls (IHC: n = 14, MS: n = 5) from one cohort. Serum p-tau 181 and p-tau 217 were higher in AD and ALS patients compared to disease controls. IHC and MS analyses revealed the presence of p-tau 181 and 217 in muscle biopsies from both ALS cases and disease controls, with ALS samples showing increased p-tau reactivity in atrophic muscle fibres. Blood p-tau species could potentially be used to diagnose both ALS and AD.

Phosphorylated (p)-tau is a well-established cerebrospinal fluid (CSF) diagnostic biomarker for Alzheimer's disease (AD)[1–3]. To date, several p-tau species, of which p-tau 181 and p-tau 217 are the most studied, have been shown to be elevated in CSF of AD patients compared to non-AD subjects[3–7]. In recent years, the increase of ultrasensitive and commercially available assays has led to the widespread measurement

of p-tau 181 and p-tau 217 in blood, with a comparable accuracy for the latter in blood and CSF[5–7].

In detail, p-tau 217 and p-tau 181 are biomarkers of initial AD neuropathological changes (Core 1 T1 category according to the Revised criteria for diagnosis and staging of AD)[2]. Indeed the two peptide levels become abnormal early, around the same time as

[1]Department of Neurology, Martin-Luther-University Halle-Wittenberg, Halle (Saale), Germany. [2]Department of Neurology and Laboratory of Neuroscience, IRCCS Istituto Auxologico Italiano, Milan, Italy. [3]Department of Medical Biotechnology and Translational Medicine, Università degli Studi di Milano, Milan, Italy. [4]Institute of Cell Biology and Neurobiology, Charité, Universitätsmedizin Berlin, Berlin, Germany. [5]Division for Neurodegenerative Diseases, Department of Neurology, Mannheim Center for Translational Medicine, University Medicine Mannheim, University of Heidelberg, Mannheim, Germany. [6]Department of Psychiatry and Psychotherapy, University Medical Center Goettingen (UMG), Georg-August University, Goettingen, Germany. [7]Department of Neurology, Ulm University Hospital, Ulm, Germany. [8]German Center for Neurodegenerative Diseases (DZNE e.V.), Ulm, Germany. [9]Department of Pathophysiology and Transplantation, "Dino Ferrari" Center, Università degli Studi di Milano, Milan, Italy. [10]Department of Geriatric Psychiatry, Central Institute of Mental Health, Medical Faculty Mannheim, University of Heidelberg, Mannheim, Germany. [11]These authors contributed equally: Samir Abu-Rumeileh, Leila Scholle. ✉e-mail: markus.otto@uk-halle.de

amyloid positron emission tomography (PET) and before tau PET, and possibly reflect the secretion of phosphorylated mid-region tau fragments in response to amyloid plaques or to soluble amyloid-β (Aβ) species, thus linking Aβ proteinopathy to early tau proteinopathy[2]. Considering their cost-effective and less invasive nature, blood p-tau 181 and p-tau 217 have recently been proposed as candidate screening tests for AD pathology in the general population[4,6,8]. Therefore, the specificity of assays for p-tau peptides is of utmost interest.

In this regard, recent studies showed high levels of blood p-tau 181 also in amyotrophic lateral sclerosis (ALS) compared to controls, despite normal CSF p-tau 181 concentrations, raising questions about the specificity of the blood marker[9–11]. Interestingly, the authors pointed to a lack of association between blood p-tau 181 and AD neuropathological changes in ALS, but rather suggested lower motor neuron (LMN) damage as a possible peripheral source of blood p-tau elevation by analysing clinical and electromyographic indices of LMN impairment and p-tau expression in brain and spinal cord samples[9,10,12]. Nevertheless, these data are limited to three cohort studies with no large multicentre analysis and no investigation of muscle tissue as an alternative peripheral source of p-tau. Furthermore, there are no data in the literature on the distribution of blood p-tau 217 in ALS.

In the present study, we aimed to i) perform a multicentre cohort study on the distribution of serum p-tau 181 and p-tau 217 levels in ALS. Although there are no major differential diagnostic concerns between ALS and AD, we decided to compare ALS with AD cases, as in previous publications[9,10], to further investigate the biological specificity of blood p-tau species. As further diagnostic groups, we included non-neurodegenerative disease controls and healthy controls. Serum total tau (t-tau), CSF AD biomarkers, serum and CSF neurofilament proteins were also measured in all diagnostic groups. ii) Furthermore, we also investigated the possible associations between clinic-demographic variables and blood p-tau 181 and p-tau 217 levels in a well-characterised group of ALS cases. iii) We searched for other potential peripheral sources of blood p-tau elevation in ALS patients by analysing ALS muscle biopsies using mass spectrometry (MS) and immunohistochemistry (IHC).

## Results

### Multicentre biomarker cohort analysis

**Serum and CSF biomarker distribution in the diagnostic groups.** Demographic characteristics and biomarker distribution of the whole sample are reported in Table 1, whereas the same data referring to the individual cohorts are described in Supplementary Tables 1–5. In the whole sample, patients with AD ($n = 111$) were older than those with ALS ($n = 152$, $p < 0.001$) and disease controls ($n = 99$, $p < 0.001$), whereas the latter groups did not differ in age. Healthy controls ($n = 23$) were significantly younger than all other groups (all $p < 0.001$). There was also a significant difference in sex distribution between all groups ($p < 0.001$) (Table 1). Serum p-tau 181 and p-tau 217 were significantly positively associated with age in the whole cohort, the control group (disease controls and healthy controls, $n = 122$) and other diagnostic groups; furthermore, there was an effect of sex on biomarker levels (Supplementary Results and Supplementary Fig. 1). The associations between age or sex and serum t-tau are reported in Supplementary Results. Taking these findings into account, biomarker comparisons between ALS, AD and disease controls have been performed after age- and sex-adjustment. The same analyses including also healthy controls are reported in Supplementary Results and Supplementary Table 1.

In the multicentre cohort, serum p-tau 181 levels were higher in patients with AD ($n = 111$) and ALS ($n = 152$) compared to disease controls ($n = 99$, both $p < 0.001$) with no differences between patients with AD and ALS (Table 1, Fig. 1A, Supplementary Fig. 2). These results were maintained in the Milan, Mannheim 1 and Mannheim 2 cohorts. In the Halle cohort, ALS patients ($n = 63$) showed higher biomarker concentrations than disease controls ($n = 40$, $p < 0.001$) and a non-significant trend toward higher levels compared to AD subjects ($n = 66$, $p = 0.077$) (Fig. 1B, Supplementary Tables 2–5). After age- and sex adjustment, we confirmed the same results (AD, $n = 111$ vs. disease controls, $n = 99$: $\beta = 0.444$, $p < 0.001$; ALS, $n = 152$ vs. disease controls, $n = 99$: $\beta = 0.442$, $p < 0.001$).

In the multicentre cohort, patients with AD ($n = 111$) and ALS ($n = 152$) yielded higher serum p-tau 217 values compared to disease controls ($n = 99$, both $p < 0.001$). Moreover, serum p-tau 217 concentrations were more elevated in AD compared to ALS ($p < 0.001$) (Table 1, Fig. 1C, Supplementary Fig. 2). These results have been

**Table 1 | Demographic characteristics and biomarker distribution in the whole sample**

| Whole cohort | ALS | AD | Disease controls | *p*-value |
|---|---|---|---|---|
| N with serum samples | 152 (127 with both serum and CSF samples) | 111 | 99 | |
| Age (years) mean ± SD | 62.34 ± 11.63 | 74.58 ± 7.39 | 61.11 ± 13.84 | <0.001 |
| Female (%) | 32.9 | 58.6 | 50.5 | 0.001 |
| Serum p-tau 181 (pg/ml) Median (IQR) | 3.0 (1.6–4.7) | 2.8 (1.8–3.6) | 1.1 (0.8–1.7) | <0.001 |
| Serum p-tau 217 (pg/ml) Median (IQR) | 0.37 (0.19–0.55) | 0.73 (0.43–0.92) | 0.14 (0.11–0.20) | <0.001 |
| Serum t-tau (pg/ml) Median (IQR) | 0.139 (0.078–0.202) | 0.245 (0.162–0.358) | 0.116 (0.058–0.206) | <0.001 |
| Serum NfL (pg/ml) Median (IQR) | 120.0 (70.5–184.3) | 45.0 (33.0–60.7) | 22.2 (13.0–34.8) | <0.001 |
| CSF p-tau 181 (pg/ml) Median (IQR) | 29.5 (20.8–39.5) | 116.0 (86.4–146.0) | 28.5 (23.0–39.6) | <0.001 |
| CSF NfH (pg/ml) Median (IQR) | 8504 (4351–13,877) | 1671 (1267–2178) | 1048 (754–1497) | <0.001 |

Age, sex and biomarkers (serum p-tau 181, serum p-tau 217, serum t-tau, serum NfL, CSF p-tau 181, CSF NfH) in the three diagnostic groups from all cohorts are displayed as mean ± standard deviation (SD), median and interquartile range (IQR) or as percentage. Depending on the type and distribution of the data, two-sided *p*-values of Kruskal–Wallis, ANOVA or Chi-test are reported. Dunn post-hoc tests (adjustments for multiple comparisons) to compare neurofilament levels between diagnostic groups: serum NfL: ALS vs AD $p < 0.001$, ALS vs controls $p < 0.001$, AD vs controls $p < 0.001$; CSF NfH: ALS vs AD $p < 0.001$, ALS vs controls $p < 0.001$, AD vs controls $p < 0.001$.
*AD* Alzheimer's disease, *ALS* amyotrophic lateral sclerosis, *CSF* cerebrospinal fluid, *IQR* interquartile range, *N* number of cases, *NfH* neurofilament heavy chain protein, *NfL* neurofilament light chain protein, *p-tau* phosphorylated tau protein, *SD* standard deviation, *t-tau* total tau protein.

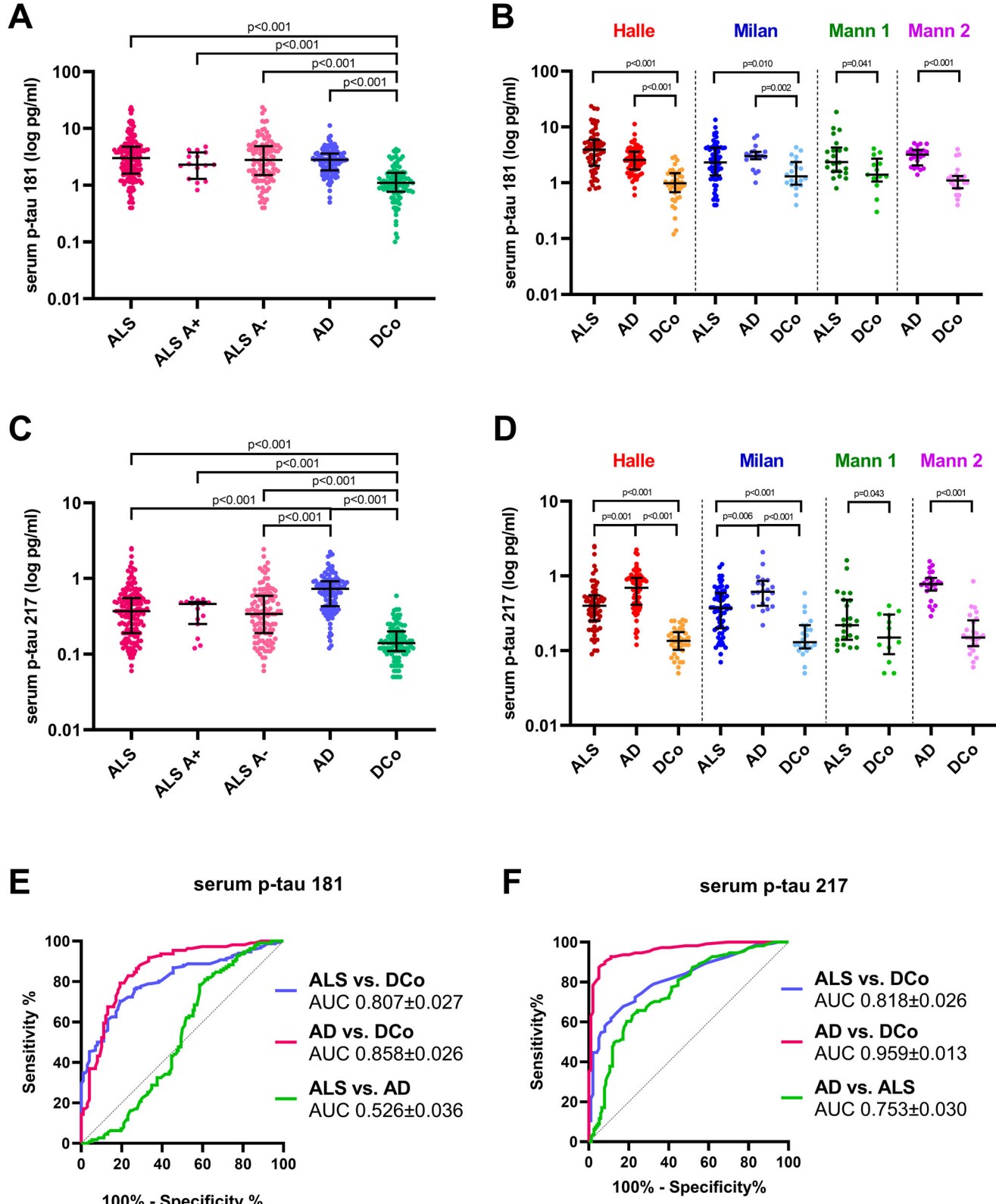

replicated in the individual cohorts (Fig. 1D, Supplementary Tables 2–5) and even after age- and sex-adjustment (AD, $n = 111$ vs. disease controls, $n = 99$: $\beta = 0.720$, $p < 0.001$; ALS, $n = 152$, vs. disease controls, $n = 99$: $\beta = 0.477$, $p < 0.001$; AD, $n = 111$ vs. ALS, $n = 152$: $\beta = 0.368$, $p < 0.001$).

Serum t-tau levels were higher in AD subjects ($n = 104$) compared to disease controls ($n = 90$, $p < 0.001$) and ALS patients ($n = 141$,

$p < 0.001$), with no difference between the latter two groups (Table 1, Fig. 2A, Supplementary Fig. 2). The same results have been confirmed in the individual cohorts (Fig. 2B, Supplementary Tables 2–5) and even after age- and sex-adjustment (AD, $n = 104$ vs. disease controls, $n = 90$: $\beta = 0.427$, $p < 0.001$; AD, $n = 104$ vs. ALS, $n = 141$: $\beta = 0.373$, $p < 0.001$).

As expected, CSF p-tau 181 was higher in AD patients ($n = 111$) compared to ALS patients ($n = 152$) and disease controls ($n = 99$, both

**Fig. 1 | Distribution of serum p-tau 181 and serum p-tau 217 in the diagnostic groups and biomarker diagnostic accuracies. A** Distribution of serum p-tau 181 in the total cohort of ALS [n = 152, including 16 cases with (A+) and 111 cases without AD co-pathology (A−)], AD cases (n = 111) and disease controls (DCo, n = 99).
**B** Serum p-tau 181 in the diagnostic groups stratified according to the individual cohorts (Halle cohort: 63 ALS, 66 AD and 40 disease controls; Milan cohort: 67 ALS, 20 AD and 22 disease controls; Mannheim 1 cohort: 22 ALS and 12 disease controls; Mannheim 2 cohort: 25 AD and 25 controls). **C** Distribution of serum p-tau 217 in the total cohort of ALS [n = 152, including 16 cases with (A+) and 111 cases without AD co-pathology (A−)], AD cases (n = 111) and DCo (n = 99). **D** Serum p-tau 217 in the diagnostic groups stratified according to the individual cohorts (Halle cohort: 63 ALS, 66 AD and 40 disease controls; Milan cohort: 67 ALS, 20 AD and 22 disease controls; Mannheim 1 cohort: 22 ALS and 12 disease controls; Mannheim 2 cohort: 25 AD and 25 controls). Biomarker levels are reported on a logarithmic scale. Dots

represent single data points. Horizontal lines represent the median values, the lower and upper lines correspond to the first and third quartiles, and the vertical line is the interquartile range. Biomarker differences between groups were assessed by Mann-Whitney or Kruskal–Wallis followed by Dunn's post hoc test (adjustment for multiple comparisons). Two-sided p-values are reported. **E** Receiver Operating Characteristic (ROC) curve of serum p-tau 181 in the distinction between ALS, AD and DCo. **F** Receiver Operating Characteristic (ROC) curve of serum p-tau 217 in the distinction between ALS, AD and DCo. Areas under the curve and standard deviation are reported. AD Alzheimer's disease, ALS amyotrophic lateral sclerosis, ALS A+ amyotrophic lateral sclerosis with AD co-pathology, ALS A- amyotrophic lateral sclerosis without AD co-pathology, AUC area under the curve, DCo disease controls, Mann 1 Mannheim 1 cohort, Mann 2 Mannheim 2 cohort, p-tau phosphorylated tau protein. Source data are provided as a Source Data file.

p < 0.001), with no difference between ALS and disease controls in the overall cohort as well as in the individual cohorts (Table 1, Supplementary Tables 2–5). As expected, ALS cases (n = 152) showed the highest serum neurofilament light chain protein (NfL) concentrations among diagnostic groups, followed by AD patients (n = 111) and then disease controls (n = 99) (Table 1, Fig. 2C, D, Supplementary Tables 2–5, Supplementary Fig. 2). The same distribution was found for CSF neurofilament heavy chain protein (NfH) (Table 1, Fig. 3E, F, Supplementary Tables 2–5, Supplementary Fig. 2).

CSF p-tau 181 and serum p-tau 181 were associated in AD (n = 111, Spearman's r = 0.304, p = 0.001) but not in ALS patients (n = 152). Serum p-tau 181 and p-tau 217 were strongly correlated in ALS (n = 152, r = 0.871, p < 0.001) and AD (n = 111, r = 0.770, p < 0.001). Furthermore, there were no significant associations between any of CSF/serum p-tau species or serum t-tau and CSF NfH or serum NfL in ALS. Other biomarker correlations are reported in Supplementary Results.

In the whole sample (n = 362), serum p-tau 181 showed a moderate diagnostic value in discriminating ALS [Area under the curve (AUC) ± standard deviation 0.807 ± 0.027] or AD (AUC 0.858 ± 0.026) from disease controls but was not able to differentiate AD from ALS (AUC 0.526 ± 0.036). Serum p-tau 217 showed a high performance in discriminating AD from disease controls (AUC 0.959 ± 0.013), but its accuracy was only moderate in the comparisons ALS vs. disease controls (AUC 0.818 ± 0.026) or ALS vs. AD (AUC 0.753 ± 0.030) (Fig. 1E, F). ROC analyses of serum t-tau are reported in Supplementary Results.

ALS patients with (CSF A+ profile, n = 16) and without AD co-pathology (A-, n = 111) showed similar serum p-tau 181 and p-tau 217 concentrations. However, the latter group had lower p-tau 217 levels (p < 0.001) and similar p-tau 181 levels compared to AD patients (n = 111) (Fig. 1A, C). AD patients had higher serum t-tau levels compared to disease controls (p < 0.001) and ALS without AD co-pathology (p < 0.001) but similar concentrations to ALS cases with AD co-pathology (Fig. 2A).

**Associations between serum tau biomarkers and clinical and laboratory variables in ALS patients.** Clinical and laboratory data of ALS patients from Halle and Milan (n = 130) are reported in detail in Table 2. Here, possible associations between serum p-tau 181, p-tau 217 and t-tau and clinical or laboratory variables were investigated. Disease duration at blood sampling was longer in ALS patients from Halle (n = 63) than in those from Milan (n = 67) (mean 32 vs. 13 months, p < 0.001). 3.1%, 50.0%, 35.4% and 11.5% of ALS subjects were in the disease stages one, two, three and four of the King's staging system, respectively. As expected, 59.3%, 19.2% and 70.6% of ALS cases showed normal values of blood CK (n = 118), troponin T (n = 26) and myoglobin (n = 34).

In ALS, serum p-tau181 was positively strongly associated with blood troponin T (n = 26, r = 0.647, p = 0.008) and weakly positively

with disease duration at blood sampling (n = 130, r = 0.211, p = 0.016) and negatively with ALSFRS-R score (n = 88, r = −0.279, p = 0.008). There were no associations of the biomarker with BMI, site of symptom onset, disease progression rate, King's stage, blood CK and blood myoglobin. Similarly, we found a positive association between serum p-tau 217 and blood troponin T (n = 26, r = 0.769, p < 0.001) but not with other of the above-mentioned variables.

Clinical phenotypes with predominant LMN involvement (LMN-predominant ALS, n = 22) showed higher serum p-tau 181 levels than bulbar ALS (n = 16, p = 0.007) and phenotypes with predominant upper motor neuron involvement (UMN-predominant ALS, n = 9, p = 0.002). Also classic ALS (n = 83) had higher serum p-tau 181 concentrations than UMN-predominant ALS (n = 9, p = 0.038). We found a similar distribution for p-tau 217 in the single-comparison analysis (p = 0.024), which remained a non-significant trend after correction for multiple comparisons. Serum t-tau was not associated with any of the above-mentioned variables.

## P-tau 181 and p-tau 217 immunoreactivity in muscle biopsies
In disease controls (n = 14) and ALS (n = 13) muscle biopsies, both p-tau 181 and p-tau 217 immunoreactivity was predominantly localised to the myonuclei, sometimes extending into the perinuclear regions (Fig. 3). Nuclear reactivity appeared to increase with age. No relevant sarcoplasmic reactivity was observed in any of the disease controls studied. However, in all ALS samples examined, there was an increased sarcoplasmic reactivity to p-tau181 and p-tau217 in atrophic muscle fibres (Fig. 3), whereas normal or hypertrophic fibres did not show increased sarcoplasmatic reactivity. In addition to muscle cells, both p-tau 181 and p-tau 217 reactivity was also observed in the walls of intramuscular vessels (tunica media). While axons of intramuscular nerves stained strongly for p-tau 181, only low reactivity was seen for p-tau 217.

## Mass spectrometry-based analysis of tau phosphorylation in muscle biopsies
Tau phosphorylation in the muscles of ALS patients (n = 5) and disease controls (n = 5) was analysed by MS-based phosphoproteomics. A total of 14 phosphorylation sites were identified in muscle biopsies with a posterior error probability less than 0.02 and a delta score greater than 40 compared to the next best spectrum match (Table 3). The presence of tau phosphorylations at T181 (p-tau 181) and T217 (p-tau 217) was confirmed in protein extracts of all analysed ALS and disease control muscle biopsies (Fig. 4A, B). In addition, several other tau phosphorylation sites were identified in muscle protein extracts (Fig. 4C). Spectra for these sites are displayed in Supplementary fig. 3. The identified sites were found to cluster in two protein regions, spanning from T171 to S235 and S365 to S373. These regions correspond to the proline-rich region and junction from repeat 4 in the microtubule-binding region to the C-terminal domain (Fig. 4C). Two phosphorylation sites at S437 and S438 of the full-length protein

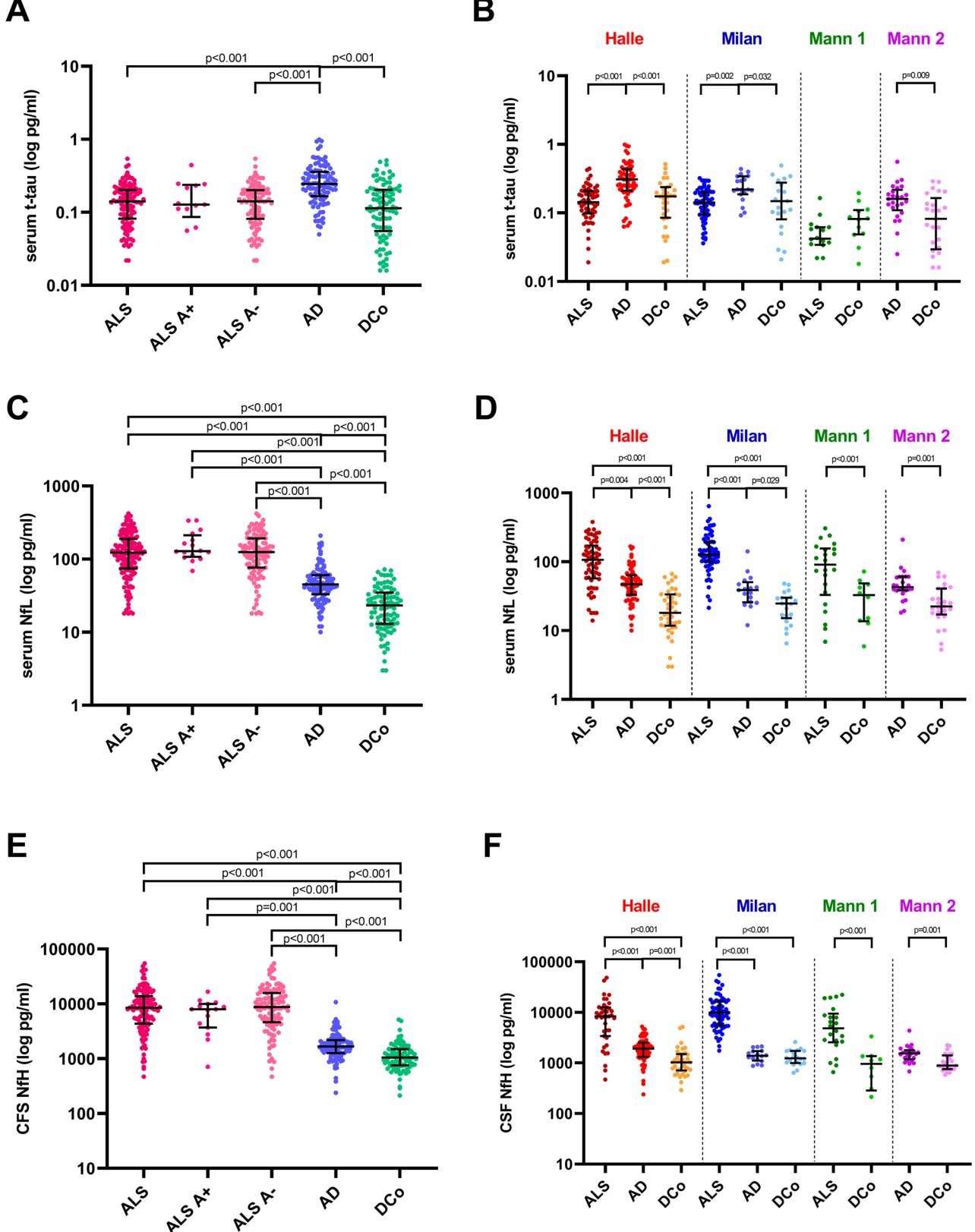

were identified with the phosphoproteomic approach in a protein region encoded by exon 6 that is not expressed in human central nervous system (CNS) tau[13].

No difference in total tau levels (tau abundance) in muscle biopsies between ALS and disease controls was detected by quantitative MS analysis (Supplementary Fig. 4).

## Discussion

In the present study, we have confirmed, in a large multicentre cohort, that serum p-tau 181 is elevated in both ALS and AD cases compared to disease controls, and we have shown, that this is also true for serum p-tau 217. Most importantly, we documented the presence of p-tau 181 and p-tau 217 in muscle biopsies from ALS cases using mass

**Fig. 2 | Distribution of serum t-tau, serum NfL and CSF NfH in the diagnostic groups. A** Distribution of serum t-tau in the total cohort of ALS [$n = 141$, including 14 cases with (A+) and 104 cases without AD co-pathology (A-)], AD cases ($n = 104$) and disease controls (DCo, $n = 90$). **B** Serum t-tau in the diagnostic groups stratified according to the individual cohorts (Halle cohort: 58 ALS, 60 AD and 32 disease controls; Milan cohort: 66 ALS, 19 AD and 21 disease controls; Mannheim 1 cohort: 17 ALS and 12 disease controls; Mannheim 2 cohort: 25 AD and 25 controls).
**C** Distribution of serum NfL in the total cohort of ALS [n = 152, including cases with (A+) and cases without AD co-pathology (A-)], AD cases ($n = 111$) and DCo ($n = 99$). **D** Serum NfL in the diagnostic groups stratified according to the individual cohorts (Halle cohort: 63 ALS, 66 AD and 40 disease controls; Milan cohort: 67 ALS, 20 AD and 22 disease controls; Mannheim 1 cohort: 22 ALS and 12 disease controls; Mannheim 2 cohort: 25 AD and 25 controls). **E** Distribution of CSF NfH in the total cohort of ALS [n = 152, including cases with (A+) and cases without AD co-pathology (A-)], AD cases ($n = 111$) and DCo ($n = 99$). **F** CSF NfH in the diagnostic groups

stratified according to the individual cohorts (Halle cohort: 63 ALS, 66 AD and 40 disease controls; Milan cohort: 67 ALS, 20 AD and 22 disease controls; Mannheim 1 cohort: 22 ALS and 12 disease controls; Mannheim 2 cohort: 25 AD and 25 controls). Biomarker levels are reported on a logarithmic scale. Dots represent single data points. Horizontal lines represent the median values, the lower and upper lines correspond to the first and third quartiles, and the vertical line is the interquartile range. Biomarker differences between groups were assessed by Mann–Whitney or Kruskal–Wallis followed by Dunn's post hoc test (adjustment for multiple comparisons). Two-sided *p*-values are reported. AD Alzheimer's disease, ALS amyotrophic lateral sclerosis, ALS A+ amyotrophic lateral sclerosis with AD copathology, ALS A- amyotrophic lateral sclerosis without AD copathology, AUC area under the curve, CSF cerebrospinal fluid, DCo disease controls, Mann 1 Mannheim 1 cohort, Mann 2 Mannheim 2 cohort, NfH neurofilament heavy chain protein, NfL neurofilament light chain protein, t-tau total tau protein. Source data are provided as a Source Data file.

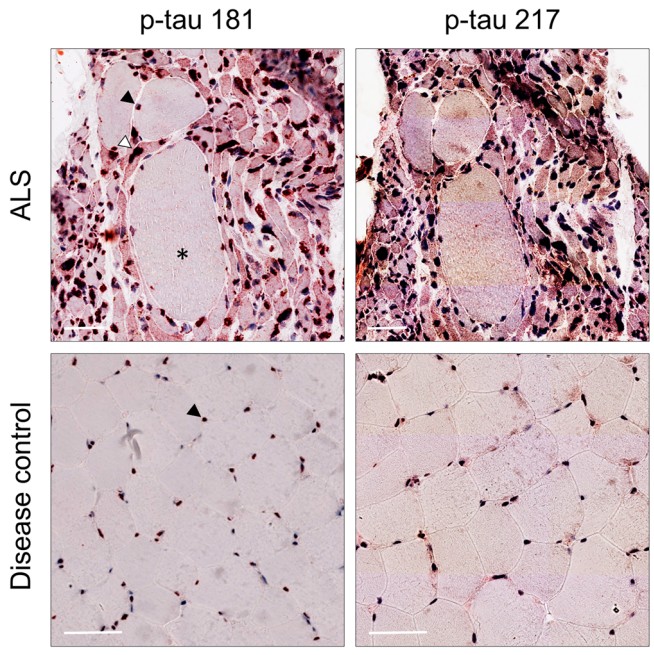

p-tau 181        p-tau 217

ALS

Disease control

**Fig. 3 | P-tau 181 and p-tau 217 immunoreactivity in muscle biopsies.** Immunohistochemical findings from a disease control (subject no. 8, male, 45 years old, diagnose: myalgia and cramps, disease duration 24 months, biopsy from biceps brachii, further details in Supplementary Table 6) and a patient with ALS (subject no. 4, male, 75 years old, disease duration 29 months, definite ALS according to the revised El Escorial criteria, King's stage 4, biopsy from vastus lateralis, further details in Supplementary Table 6) are shown. Muscle biopsies from disease controls ($n = 14$) showed normal-sized, predominantly hexagonal fibres with random variations in muscle fibre size and shape. Conversely, biopsies from patients with ALS ($n = 13$) showed neurogenic features, including grouped or even fascicular atrophy, with small-angled fibres (example of atrophic fibre marked with white arrowhead) and nuclear bags, often accompanied by compensatory hypertrophy of unaffected fibres (black asterisk). Both ALS ($n = 13$) and disease control ($n = 14$) muscle biopsies showed p-tau 181 and p-tau 217 immunoreactivity (scale bar: 50 μm) predominantly localised to the myonuclei (black arrowheads), sometimes extending into the perinuclear regions. In all ALS samples, we found increased sarcoplasmic reactivity to p-tau181 and p-tau 217 in atrophic muscle fibres (white arrowhead), whereas normal or hypertrophic fibres did not show increased sarcoplasmic reactivity (black asterisk).

**Table 2 | Demographic, clinical and laboratory data of ALS patients from Halle and Milan ($n = 130$)**

| | |
|---|---|
| **Age (years) at blood collection** mean ± SD | 63.24 ± 10.50 |
| **Female** % | 41 (31.6) |
| **BMI** mean ± SD | 24.12 ± 3.73 |
| **Site of symptom onset** N (%) | |
| Bulbar | 27 (20.8) |
| Spinal: upper limbs, trunk, lower limbs | 98 (75.4): 49 (37.7), 3 (2.3), 46 (35.4) |
| Generalised | 4 (3.1) |
| Cognitive | 1 (0.7) |
| **Disease duration at blood collection (months)** mean ± SD | 21.82 ± 20.33 |
| **Clinical phenotype** N(%) | |
| Classic | 83 (63.9) |
| Bulbar | 16 (12.3) |
| UMN-predominant | 9 (6.9) |
| LMN-predominant | 22 (16.9) |
| **Diagnosis according to revised El Escorial criteria** | |
| Definite | 31 (23.8) |
| Probable | 30 (23.1) |
| Probable laboratory supported | 46 (35.4) |
| Possible | 23 (17.7) |
| **King's staging system** | |
| 1 | 4 (3.1) |
| 2 | 65 (50.0) |
| 3 | 46 (35.4) |
| 4 | 15 (11.5) |
| **ALSFRS-R score** mean ± SD | 38.54 ± 8.17 |
| **Disease progression rate at blood sampling** mean ± SD | 0.71 ± 0.62 |
| **ECAS total score** mean ± SD | 100.57 ± 17.58 |
| **Blood Creatine Kinase (U/L)** Median (IQR) (normal values < 171 U/L) % of cases with normal values | 133.0 (65.9-253.4) 59.3 |
| **Blood Troponin T (ng/L)** Median (IQR) (normal values < 14 ng/L) % of cases with normal values | 51.0 (16.9-74.3) 19.2 |
| **Blood Myoglobin (µg/L)** Median (IQR) (normal values 25-72 µg/L) % of cases with normal values | 43.0 (30.6-89.5) 70.6 |

*ALSFRS-R* ALS Functional Rating Scale-Revised, *BMI* body mass index, *ECAS* Edinburgh Cognitive and Behavioural ALS Screen, *IQR* interquartile range, *LMN* lower motor neuron, *SD* standard deviation, *UMN* upper motor neuron.

spectrometry and immunohistochemistry, providing evidence that striated muscle tissue may be an additional peripheral source of blood p-tau.

First, our data are consistent with previous data on blood p-tau 181 distribution in ALS and AD[9,10] by providing an external validation in

**Table 3 | Tau phosphorylation sites identified in muscle biopsies by mass spectrometry**

| Position in can. Tau | Isoforms | Positions in Tau-isoforms | Can. Tau | 2N4R Tau | Amino acid | Localisation probability | PEP | Score | Delta score | Mass error [ppm] | Intensity | Phospho (STY) Probabilities |
|---|---|---|---|---|---|---|---|---|---|---|---|---|
| 437 | Tau (can.); Tau-G | 437; 437 | x | – | S | 0.49999 | 3.50E-19 | 91.767 | 74.609 | 0.1029 | 3.35E+06 | HPTPGSSDPLIQPSSPAVCPEPPS(0.5)S(0.5)PK |
| 438 | Tau (can.); Tau-G | 438; 438 | x | – | S | 0.49999 | 3.50E-19 | 91.767 | 74.609 | 0.1029 | 3.35E+06 | HPTPGSSDPLIQPSSPAVCPEPPS(0.5)S(0.5)PK |
| 498 | Tau (can.); 2N3R; 2N4R; ON3R; ON4R; Tau-A; 1N3R; 1N4R | 498; 181; 181; 123; 123; 87; 152; 152 | x | x | T | 0.999742 | 9.69E-11 | 126.4 | 114.23 | 0.5389 | 3.44E+08 | TPPAPKT(1)PPSSGEPPK |
| 516 | Tau (can.); 2N3R; 2N4R; Tau-G; ON3R; ON4R; Tau-A; 1N3R; 1N4R | 516; 199; 199; 534; 141; 141; 105; 170; 170 | x | x | S | 0.997557 | 5.85E-34 | 188.92 | 188.92 | -0.0025 | 4.48E+07 | SGYS(0.002)S(0.998)PGSPGTPGSR |
| 519 | Tau (can.); 2N3R; 2N4R; Tau-G; ON3R; ON4R; Tau-A; 1N3R; 1N4R | 519; 202; 202; 537; 144; 144; 108; 173; 173 | x | x | S | 0.996594 | 2.07E-33 | 185.67 | 166.46 | 0.1095 | 2.41E+07 | SGYSSPGS(0.997)PGT(0.003)PGSR |
| 531 | Tau (can.); 2N3R; 2N4R; Tau-G; ON3R; ON4R; Tau-A; 1N3R; 1N4R | 531; 214; 214; 549; 156; 156; 120; 185; 185 | x | x | S | 0.983609 | 0.0005762 | 73.156 | 57.018 | 0.2188 | 1.06E+06 | T(0.001)PS(0.984)LPT(0.016)PPTREPK |
| 534 | Tau (can.); 2N3R; 2N4R; Tau-G; ON3R; ON4R; Tau-A; 1N3R; 1N4R | 534; 217; 217; 552; 159; 159; 123; 188; 188 | x | x | T | 0.999984 | 3.56E-53 | 190.44 | 128.08 | 0.0019 | 1.44E+07 | TPSLPT(1)PPTREPK |
| 548 | Tau (can.); 2N3R; 2N4R; Tau-G; ON3R; ON4R; Tau-A; 1N3R; 1N4R | 548; 231; 231; 566; 173; 173; 137; 202; 202 | x | x | T | 0.999996 | 6.67E-08 | 141.22 | 130.08 | -0.3566 | 1.85E+08 | VAVVRT(1)PPKS(0.974)PS(0.022)S(0.004)AK |
| 552 | Tau (can.); 2N3R; 2N4R; Tau-G; ON3R; ON4R; Tau-A; 1N3R; 1N4R | 552; 235; 235; 570; 177; 177; 141; 206; 206 | x | x | S | 0.976763 | 0.000152621 | 103.9 | 82.593 | -0.3676 | 1.85E+07 | VAVVRT(1)PPKS(0.977)PS(0.02)S(0.004)AK |
| 579 | Tau (can.); 2N3R; 2N4R; Tau-G; ON3R; ON4R; Tau-A; 1N3R; 1N4R | 579; 262; 262; 597; 204; 204; 168; 233; 233 | x | x | S | 0.991753 | 0.0116387 | 109.83 | 42.868 | 0.1405 | 2.58E+06 | IGS(0.992)T(0.008)ENLK |
| 673 | Tau (can.); 2N3R; 2N4R; Tau-G; ON3R; ON4R; Tau-A; 1N3R; 1N4R | 673; 325; 356; 691; 307; 298; 231; 296; 327 | x | x | S | 0.999979 | 2.77E-07 | 101.48 | 101.48 | -0.3446 | 3.03E+06 | IGS(1)LDNITHVPGGGNKK |
| 713 | Tau (can.); 2N3R; 2N4R; Tau-G; ON3R; ON4R; Tau-A; 1N3R; 1N4R | 713; 365; 396; 731; 307; 338; 271; 336; 367 | x | x | S | 0.999971 | 1.25E-43 | 148.85 | 140.95 | -0.1765 | 2.37E+08 | TDHGAEIVYKS(1)PVVS(0.024)GDT(0.537)S(0.44)PR |
| 717 | Tau (can.); 2N3R; 2N4R; Tau-G; ON3R; ON4R; Tau-A; 1N3R; 1N4R | 717; 369; 400; 735; 311; 342; 275; 340; 371 | x | x | S | 0.999931 | 1.57E-24 | 148.85 | 140.95 | 0.2650 | 6.78E+07 | SPVVS(1)GDT(0.26)S(0.74)PR |
| 721 | Tau (can.); 2N3R; 2N4R; Tau-G; ON3R; ON4R; Tau-A; 1N3R; 1N4R | 721; 373; 404; 739; 315; 346; 279; 344; 375 | x | x | S | 0.964325 | 6.13E-81 | 218.02 | 187.85 | -0.0990 | 1.63E+09 | SPVVSGDT(0.036)S(0.964)PR |

can. canonical isoform 1, *PEP* posterior error probability.

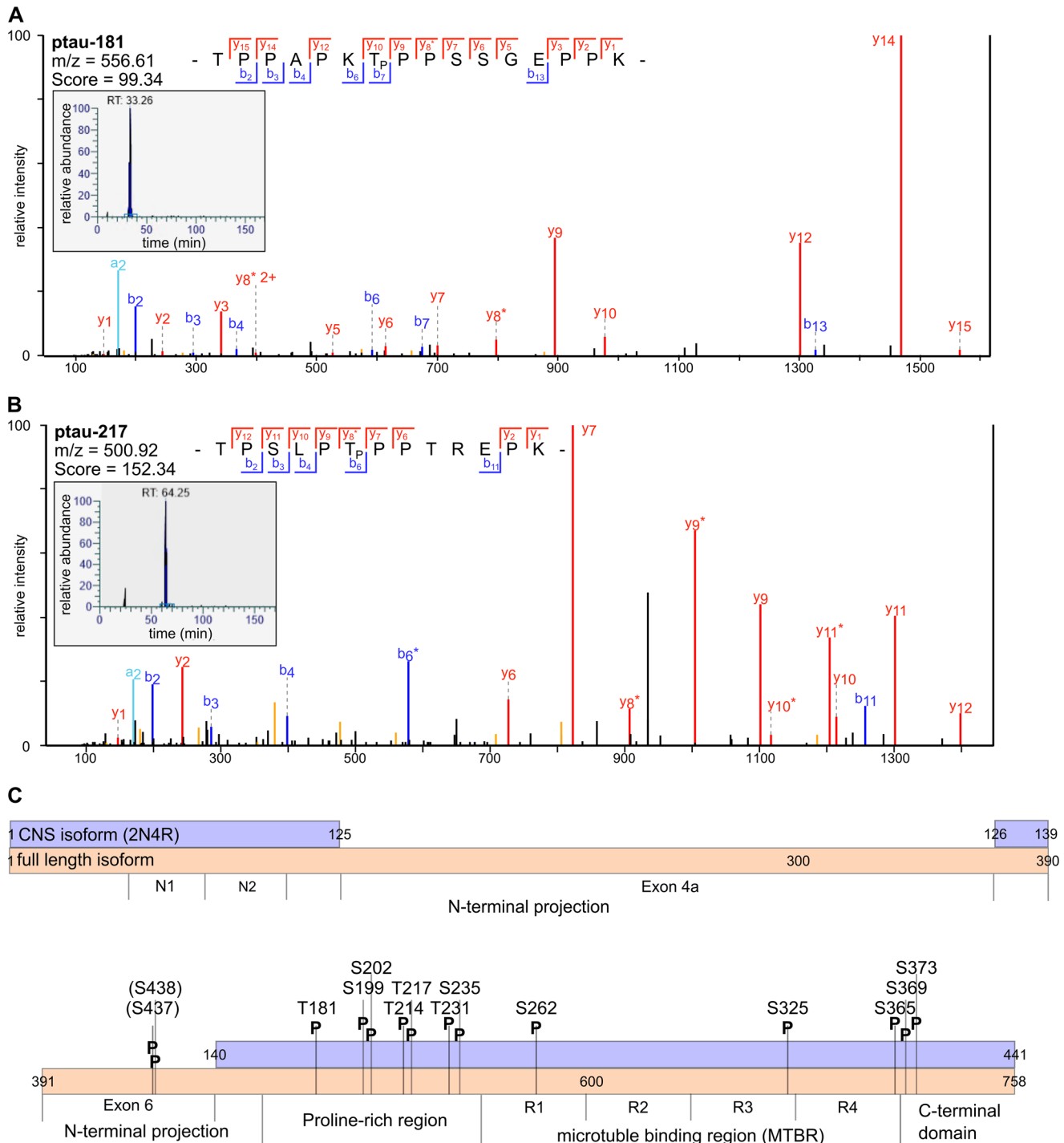

**Fig. 4 | Mass spectrometry-based analysis of tau phosphorylation in muscle biopsies.** Exemplary spectra and for the identification of p-tau 181 (**A**) and p-tau 217 (**B**) in muscle extracts are shown. Both phosphosites were identified in all analysed ALS patients (*n* = 5) and disease controls (*n* = 5). **C** Additional phosphorylation sites were identified in all samples. Two sites, S437 and S438, were identified in a protein region that is not expressed in human CNS tau. Protein regions shown in purple-blue are expressed in adult CNS and other tissues. Protein regions shown in orange, but not in purple-blue, are not expressed in adult CNS. Positions are indicated as the site in the CNS-expressed isoform. Sites outside this isoform are given with respect to the full-length protein. CNS central nervous system, p-tau phosphorylated tau.

four different cohorts recruited in European reference centres for neurodegenerative diseases. As a new finding, we also documented significantly higher levels of blood p-tau 217 in ALS, but to a lesser extent than in AD, a result that well supports the higher accuracy of p-tau 217 for AD pathology compared to p-tau 181, as previously described[5,7]. Of note, the high serum p-tau 181 and p-tau 217 levels in ALS participants from the Halle cohort compared to other cohorts may

be explained by our finding of an association between biomarker levels and disease duration at blood sampling and the longer disease duration of ALS patients in this cohort. Indeed, Vacchiano et al. showed a longitudinal increase in blood p-tau 181 levels during the disease course in ALS patients[10]. In this respect, further studies are here needed to investigate whether blood p-tau 217 shows the same temporal dynamics.

Most interestingly, in ALS, the strong correlations between blood p-tau 181 and blood troponin T, a marker of striated and myocardial muscular damage which is usually elevated in ALS (together with other muscular markers)[14–16], the lack of correlation between CSF and serum p-tau 181 levels[9,10,12], and the higher serum marker levels in LMN-predominant ALS[10], indicate that a substantial fraction of blood p-tau 181 in ALS is probably derived from a peripheral source. Furthermore, we also found a strong association between blood p-tau 217 and blood troponin T, whereas Verde et al. found a correlation between blood CK and blood p-tau 181[12].

Notably, p-tau 181 and p-tau 217 have been proposed as "more CNS specific tau species" in blood than t-tau and other p-tau species, such as p-tau 202[17]. Indeed, it has been estimated that approximately 80% of blood t-tau is derived from peripheral sources[18]. Altogether, our current findings indicate that the hypothesis of an exclusive brain-derived source of blood p-tau181 and blood p-tau 217 does not appear to apply in ALS patients. Of note, our data suggest that the increase in serum p-tau species in ALS might be related to the elevation in p-tau and not to a general increase in t-tau levels, as t-tau concentrations in serum did not differ between ALS and disease controls in agreement with previous studies[19,20]. Here, the relatively similar or higher levels of serum p-tau species in relation to the corresponding serum t-tau values in our cohort were also in line with previous literature[21,22] and might be due to the different sensitivities of the assays.

In a biochemical-pathological translational approach, we found that our results on serum tau biomarkers were also consistent with our findings in muscle biopsies. Indeed, our MS analysis documented the presence of p-tau 181 and p-tau 217 in all ALS and disease control muscle samples and, as with for serum samples, no difference in t-tau concentrations between the two groups. Interestingly, we found two additional phosphorylation sites (S437 and S438) in a protein region that is not present in the splicing variants of human CNS tau[13] and could, therefore, be muscle-specific. Indeed, shorter tau isoforms are expressed in the CNS, lacking two additional protein regions compared to the full-length protein expressed in other tissues[13]. This would open new horizons for further studies on these species as potential muscle biomarkers.

Similarly, IHC analysis revealed p-tau 181 and p-tau 217 reactivity in both disease controls and ALS muscle biopsies, but most interestingly, all ALS samples showed significantly increased sarcoplasmic reactivity for both p-tau species in atrophic muscle fibres but not in normal or hypertrophic fibres, suggesting denervated muscle fibres as the possible peripheral source of blood p-tau 181 and 217 elevation in ALS.

In addition, the finding of increasing nuclear p-tau 181 and p-tau 217 reactivity with age in both ALS cases and disease controls is consistent with strong good correlations between age and serum p-tau 181 and 217 in our control group and may suggest an age- or senescence-related effect on the release of p-tau species from muscle tissue, which may also be independent of disease states and warrants further investigations.

To date, only very few studies have investigated the expression of tau protein and/or phosphorylated tau in muscle biopsies, either in patients with inclusion body myositis (striated muscle) or in healthy subjects (lung smooth muscle)[23–25], and our study is the first to be performed in ALS patients. Altogether, our data do not contradict previous studies on p-tau 181 in ALS, which suggested LMN dysfunction and degeneration as responsible for the biomarker increase in blood[9,10,12]. Indeed, p-tau 181 and p-tau 217 could be released from peripheral axons and the corresponding denervated muscle fibres, which are in close contact[9,10]. Similarly, although tau pathology has also been found in the motor cortex (UMN) and spinal cord (LMN) in ALS[9,26–28], its burden does not seem to correlate with blood p-tau levels[9], further reinforcing the hypothesis of the contribution of a peripheral source. Notably, pathological tau species have also been found in the blood and in other peripheral tissues (i.e. skin, nerves) of

patients with progressive supranuclear palsy, a tauopathy[29–31], further supporting the localisation of tau species outside the CNS.

Nevertheless, given the small sample size, the lack of paired blood and muscle samples and the consequent inability to make direct quantitative comparisons between groups on muscle data, our neuropathological analyses should be considered as preliminary, and future large cohort studies with ALS and AD muscle biopsies are needed to confirm our findings. It may also be of great interest to investigate tau mRNA in muscle biopsies as well as the possible pathological mechanisms and longitudinal course of peripheral tau pathology in ALS, AD and other neurological diseases.

On another issue, the lack of correlation between CSF or serum neurofilaments and serum p-tau 181 and p-tau 217 concentrations supports the notion that p-tau changes in blood are unlikely to be related to neuroaxonal damage in ALS[9,10]. It is noteworthy that blood and CSF neurofilaments have shown a prognostic, monitoring and pharmacodynamic role in ALS, in addition to a high diagnostic performance[32–36]. In contrast, the prognostic value of blood p-tau species and their longitudinal changes in ALS are poorly understood[10] and should be further investigated.

Overall, our results and those of previous studies highlight the problem, that blood p-tau concentrations measured by currently available commercial assays may be affected by the confounding quantification of species of peripheral origin. Considering the very different clinical pictures of ALS and AD, the overlap of blood p-tau 181 and 217 levels between ALS and AD may not challenge the diagnostic assessment in the symptomatic phase but might significantly affect the biomarker interpretation for screening purposes in the pre-symptomatic and early disease phases[37]. Accordingly, we found only moderate and poor performance of serum p-tau 217 and p-tau 181, respectively, in the discrimination between AD and ALS. Therefore, more specific p-tau 181 and 217 assays should be tested and validated in the future before these could be proposed as accurate screening tests for AD pathology in the general population.

The main strengths of our study are its multicentric nature and the detailed clinical characterisation of a large proportion of ALS cases, as well as the use of different biospecimens (serum, CSF and muscle biopsy) and techniques (ultrasensitive ELISAs, IHC and MS) to investigate p-tau species. On the other hand, limitations include the lack of paired biospecimens, lack of consistency in the location of muscle biopsies, and lack of information on patients' co-morbidities other than AD, which could have influenced the experimental results. In addition, the CSF biomarker cut-offs used to define the CSF-based AT(N) status were slightly different between centres, but this should not have affected our results, given their good reproducibility across cohorts. In addition, unlike previous studies[9,10], electrophysiological data as well as brain and spinal tissue samples were not available from our ALS cases. In addition, our study lacked muscle biopsies from AD cases and healthy controls.

In conclusion, the elevation of serum p-tau 181 and p-tau 217 in ALS subjects probably reflects the release of these species from denervated muscle fibres and point to the need for further studies on peripheral tau pathology in ALS and other neuromuscular diseases. On the other hand, p-tau species could potentially be used to diagnose both ALS and AD, thus calling into question the recently proposed inclusion of blood p-tau 181 and p-tau 217 as accurate screening tests for AD pathology.

## Methods
### Inclusion and ethics
The study was conducted according to the revised Declaration of Helsinki and Good Clinical Practice guidelines. Informed written consent was obtained from participants and/or their relatives. The study of biosamples and case data was approved by the ethics committees of the Martin-Luther-University Halle-Wittenberg (approval number

2021-101), IRCCS Istituto Auxologico Italiano (approval number 2023_03_21_18) and University Medicine Mannheim (approval numbers 2017-589N-MA and 2012-254N-MA). The study followed the STROBE statement guidelines.

## Multicentre biomarker cohort analysis

**Case selection.** In this retrospective multicentre cohort study, we included ALS and AD (dementia or mild cognitive impairment) cases and non-neurodegenerative disease controls from four different cohorts (total $n = 362$). The case selection was based on the availability of serum samples from subjects with the target conditions and complete information for diagnostic criteria application. No statistical method was used to predetermine sample size. Participants were recruited at the Department of Neurology, Martin-Luther-University Halle-Wittenberg, Halle (Saale), Germany (Halle cohort, $n = 169$), Department of Neurology and Laboratory of Neuroscience, IRCCS Istituto Auxologico Italiano, Milan, Italy (Milan cohort, $n = 109$), at the Division for Neurodegenerative Diseases, Neurology Department, Mannheim Center for Translational Medicine, University Medicine Mannheim, Mannheim, Germany (Mannheim cohort 1, $n = 34$), or at the Department of Geriatric Psychiatry, Central Institute of Mental Health, Mannheim, Germany (Mannheim cohort 2, $n = 50$). The Halle cohort comprised 63 ALS, 66 AD and 40 disease controls; the Milan cohort 67 ALS, 20 AD and 22 disease controls; the Mannheim 1 cohort 1 22 ALS and 12 disease controls; the Mannheim 2 cohort 25 AD and 25 controls. All AD and disease control participants as well as 127 of 152 ALS cases had available CSF samples.

ALS diagnosis was made according to the Revised El Escorial criteria[38]. The diagnosis of AD was supported by the analysis of CSF core biomarkers, according to the National Institute on Aging and Alzheimer's Association recommendations[1,2]. Non-neurodegenerative disease controls comprised subjects lacking any clinical, neuroradiological and CSF evidence of neurodegenerative disease.

To further investigate the potential impact of age on biomarkers, we also included serum samples from 23 healthy controls aged less than 50 years with normal Mini-Mental State Examination score, no history of neurological disease and normal serum NfL levels according to our in-house cut-off values. No CSF AD biomarker assessment was performed in these cases, but given their age, the occurrence of incidental AD co-pathology should be considered unlikely[39].

In ALS patients from Halle and Milan ($n = 130$), we collected, when available, clinic-demographical data and scores and laboratory results: age at blood sampling, sex, body mass index (BMI), disease duration at blood sampling (time between symptom onset and blood sampling), site of symptom onset[40], clinical phenotype[41], King's stage at blood sampling (disease staging system)[42], ALS Functional Rating Scale-Revised (ALSFRS-R) score[10,34] and disease progression rate at blood sampling[10,34], Edinburgh Cognitive and Behavioural ALS Screen (ECAS) total score at blood sampling[43], blood levels of creatine kinase (CK), troponin T and myoglobin. Clinical phenotypes comprehended classic, bulbar, UMN-predominant (including also primary lateral sclerosis) and LMN-predominant (including flail arm syndrome, flail leg syndrome and progressive muscular atrophy) ALS[10,34,44]. The disease progression rate at blood sampling was calculated as follows: (48 − ALSFRS-R score at the time of sampling)/time between symptom onset and blood sampling[10,34]. A genetic screening for the most frequent ALS-associated genes was performed. Pathogenetic mutations in *TARDBP, FUS, SOD1, ATXN2* genes as well as *C9orf72* hexanucleotide repeat expansions were found in six, one, one, one and two cases, respectively.

**Blood and CSF biomarker analyses.** Serum samples were collected in each centre, aliquoted and stored at −80 °C according to standard procedures. CSF samples were obtained by lumbar puncture (LP), centrifuged in case of blood contamination, divided into aliquots and stored in polypropylene tubes at −80 °C until analysis in each centre.

Serum p-tau 181, p-tau 217 and t-tau were measured at the Department of Neurology, Martin-Luther-University Halle-Wittenberg, Halle (Saale), by Simoa platform (Quanterix, Billerica, Massachusetts, USA) using commercially available assays (pTau-181 Advantage V2 kit, ALZpath pTau-217 Advantage PLUS kit and Tau 2.0 kit). P-tau 181 and 217 were measured in all cases whereas t-tau in the large majority of them ($n = 357$). Further details on serum p-tau measurements are reported in supplementary methods.

Serum neurofilament light chain protein (NfL) and CSF neurofilament heavy chain protein (NfH) levels were quantified by means of the Ella microfluidic system (Bio-Techne, Minneapolis, MN)[32] in Halle (Saale). CSF AD core biomarkers (A: Aβ42/Aβ40, T: p-tau 181, N: t-tau) in the Halle and Mannheim 1 cohorts were measured in Halle (Saale), whereas the other two cohorts were measured in the respective centres. All centres used an automated chemiluminescent enzyme immunoassay on the Lumipulse platform (Fujirebio, Gent, Belgium)[45]. Pathological values for AD core markers (A+, T+ and N+) were determined according to respective centre-specific in-house validated cut-offs (Halle and Mannheim 1: Aβ42/Aβ40 < 0.069, p-tau 181 > 56.5 pg/ml, t-tau >400 pg/ml; Milan: Aβ42/Aβ40 < 0.069, p-tau 181 > 56.5 pg/ml, t-tau >404 pg/ml; Mannheim 2: Aβ42/Aβ40 < 0.05, p-tau 181 > 61.0 pg/ml, t-tau >450 pg/ml).

All AD participants and disease controls showed A+T+N+ and A-T-N- CSF profiles, respectively, according to the AT(N) classification system[1]. Accordingly, ALS cases were classified as having AD-copathology in case of A + T ± CSF profiles[3,45].

All biomarker analyses were randomised, and investigators were blinded to sample allocation and diagnose. For all biomarker measurements, the coefficients of intra-assay and inter-assay variability were <10% and <15%, respectively.

## Immunohistochemical p-tau 181 and p-tau 217 staining of muscle biopsies

Muscle biopsies were collected at the Department of Neurology, Martin-Luther-University Halle-Wittenberg, Halle (Saale). Muscle biopsies from 13 ALS patients (median age at biopsy 55 years [min-max. 35-76 years]; 46% females, diagnosis according to the revised El-Escorial criteria: 8 possible ALS, 3 definite ALS, 2 probable ALS laboratory supported) were available for histological evaluation (Supplementary Table 6). We also studied 14 disease controls, who had undergone a diagnostic work-up for myalgia, cramps and exercise intolerance and who showed no pathological findings on complete clinical examination, electromyography and muscle biopsy with myohistological examination (median age at biopsy 43 years [min-max. 20−53 years], 35% females) (Supplementary table 6).

Muscle biopsies were performed as open biopsies from the muscles indicated in Supplementary Table 6, with approximately 1 cm³ of muscle tissue obtained. All histological studies were performed on frozen sections. For this purpose, the muscle biopsies were mounted on pre-cooled cork discs immediately after excision and directly frozen using isopentane cooled in a container immersed in liquid nitrogen. Once sufficiently frozen, the samples were stored in liquid nitrogen until further processing.

IHC was performed on 5 µm sections according to standard procedures of the facility. Briefly, sections were fixed in formaldehyde solution (4%), rinsed in tris-buffered saline (TBS), incubated in hydrogen peroxide solution (3%), rinsed again and treated with blocking solution (ZytoChem-Plus HRP polymer kit, Zytomed Systems, Germany). The primary antibodies against p-tau 181 (Thermo Fisher Scientific, MA, USA; AT270 #MN1050; 1:50) or p-tau 217 (Thermo Fisher Scientific, MA, USA; #44−744; 1:50) were added, respectively. This was followed by rinsing and incubation with a post-blocking reagent and species-specific HRP-conjugated secondary antibodies.

Finally, AEC substrate kit (Zytomed Systems, Germany) was used for visualisation, followed by counterstaining with hemalum solution. Histopathological evaluation of stained specimens was performed by an experienced neuropathologist (GS-D) together with two neurologists trained in muscle pathology (AK, TK), all blinded to clinical data.

Due to the very limited number of paired serum and muscle biosamples in the ALS group ($n = 2$), it was not possible to investigate the possible associations between serum biomarker levels and muscle p-tau content.

### Mass spectrometry-based phosphopeptide analysis of muscle biopsies

Protein extracts from muscle biopsies from ALS patients (5 out of the 13 cases) and disease controls (5 out of the 14 cases) (Supplementary Table 6) were subjected to proteolytic cleavage using a paramagnetic bead approach[46]. Phosphopeptides were sequentially enriched from resulting dried peptides by $TiO_2$- and Fe-NTA-based affinity chromatography as previously described[47]. The liquid chromatography–mass spectrometry / mass spectrometry (LC-MS/MS) analysis of enriched phosphopeptides and complete input controls were conducted blinded to diagnoses on an EASY-nLC 1200 system (Thermo Fisher Scientific, USA), coupled to an Orbitrap Fusion Lumos Tribrid mass spectrometer (Thermo Fisher Scientific, USA) via a FAIMS Pro DUO interface (Thermo Fisher Scientific, USA). The samples were separated by a tripartite linear 165 min gradient and analyzed utilising a data-dependent acquisition approach. The LC–MS/MS raw data were examined by MaxQuant (Version 2.4.0.0)[48]. Database search was performed against the UniProt Homo Sapiens Reviewed RefSet (03/2024, 20418 entries + isoforms) and a list of common contaminants provided by MaxQuant (07/2019, 245 entries)[49].

Stringent filtering was applied to the phosphosites identified by our mass spectrometry approach. All sites had to pass filtering for a false discovery rate of less than 0.01. The identified phosphosites were filtered based on the phosphorylated protein. Only sites identified in the protein sequence of any tau isoform were further analyzed. The site localization probability had to be >0.95 for singly phosphorylated peptides or between 0.475 and 0.525 for adjacent acceptor sites in doubly phosphorylated peptides. The overall posterior error probability had to be less than 0.02. In a final check, the identification score (delta score relative to the unmodified peptide) had to be at least 40 points greater than for the unphosphorylated counterpart. For proline-containing tau-phosphopeptides, all compensation voltage lines in all raw data were manually checked to exclude missing values due to proline isomerization. Further details on MS analyses are reported in Supplementary Methods.

All IHC and MS analyses were randomised, and investigators were blinded to sample allocation and diagnose.

### Statistical analysis and reproducibility

We used IBM SPSS Statistics V.21 (IBM), GraphPad Prism V.7 (GraphPad Software, La Jolla, California, USA), and R software V.4.0.2 (R foundation, Vienna, Austria). Depending on distribution, data were expressed as percentage, mean ± standard deviation (SD), or median and interquartile range (IQR). We adopted the $\chi^2$ test for categorical variables. For continuous variables, depending on the data distribution and number of groups, we applied the Mann–Whitney $U$ test, $t$-test, Kruskal–Wallis test (followed by Dunn–Bonferroni post hoc test) or the ANOVA (followed by Tukey's post hoc test). All reported p values were adjusted for multiple comparisons. We performed multivariate linear regression models to adjust for age and sex the differences in blood or CSF biomarkers between the groups after the transformation of the dependent variable in the natural logarithmic scale. Spearman's correlations and uni- or multivariate regression analyses were performed to test the possible associations between variables. The diagnostic accuracy of each marker was calculated by means of receiver operating characteristic (ROC) analyses. Statistical tests were two-tailed, and two-sided p values were considered statistically significant at <0.05.

### Reporting summary

Further information on research design is available in the Nature Portfolio Reporting Summary linked to this article.

## Data availability

Biomarker and mass spectrometry data generated in this study are provided in the Source Data file. Mass spectrometry data have also been deposited in the ProteomeXchange Consortium database via the PRIDE partner repository under the identifiers PXD055632 and PXD060425 for phosphopeptide analysis and protein data, respectively. All other data contain patient-related clinical information and are available on request from the corresponding author. The request will be reviewed by the investigators and the respective institutions to verify that the data transfer is in accordance with EU legislation on general data protection or is subject to intellectual property or confidentiality obligations. Source data are provided with this paper.

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

## Acknowledgements

We are thankful to Lisa Habeck, Katrin Schulz (Department of Neurology, Martin-Luther-University Halle-Wittenberg, Halle, Germany) and Angela Rosemeier (Department of Orthopedics, Martin-Luther-University Halle-Wittenberg, Halle, Germany) for their expert technical assistance and to all patients for participating in this study. We are also thankful to Ivan Valkadinov and Jasper Hesebeck-Brinckmann (Mannheim Center for Translational Medicine, University Medicine Mannheim, University of Heidelberg, Mannheim, Germany).

## Author contributions

S.A.R., L.S., and M.O. designed the study. S.A.R., L.S., A.M., H.G., A.K. and G.S.D. performed experiments. S.A.R., L.S., A.M., H.G., A.R., A.K., G.S.D., J.C., A.D.G., L.B., P.S., H.W.K., P.O., S.H., C.S., A.P., T.K., V.S., L.H., N.T., L.F., J.H.W., F.V. and M.O collected, analysed data and contributed to data interpretation. S.A.R., L.S., A.M. H.G. and M.O. wrote the manuscript. S.A.R., L.S., A.M., H.G., A.R., A.K., G.S.D., J.C., A.D.G., L.B., P.S., H.W.K., P.O., S.H., C.S., A.P., T.K., V.S., L.H., N.T., L.F., J.H.W., F.V. and M.O critically revised the manuscript.

## Funding

## Competing interests

S.A.R. received research support from the Medical Faculty of Martin-Luther-University Halle-Wittenberg (Clinician Scientist Programm No. CS22/06), unrelated to the work presented in this paper. A.M. has received advisory board honoraria and speaking fees from Hormosan and Sanofi, all unrelated to the submitted work. J.C. received research support from the German Foundation of Neurology (Deutsche Stiftung Neurologie), the Rolf-Schwiete-Stiftung, and the programme for research and education at LMU Munich (FoeFoLe-LMU), all unrelated to the work presented in the paper. L.B. received research support from the Medical Faculty of Martin-Luther-University Halle-Wittenberg (Junior Clinician Scientist Programm No. JCS24/02), unrelated to the work presented in this paper. P.O. received research support from the Cure Alzheimer Fund, ALS Association (24-SGP-691, 23-PPG-674-2), ALS Finding a Cure, the Charcot Foundation, the DZNE Innovation-to-Application programme and consulting fees from LifeArc and Funda-mental Pharma, unrelated to the work presented in this paper. V.S. received compensation for consulting services and/or speaking activities from AveXis, Cytokinetics, Italfarmaco, Liquidweb S.r.l., Novartis Pharma AG, Amylyx Pharmaceuticals, Biogen, and Zambon Biotech SA; receives or has received research supports from the Italian Ministry of Health, AriSLA, and E-Rare Joint Transnational Call, all unrelated to the work presented in this paper. V.S. is in the Editorial Board of Amyo-trophic Lateral Sclerosis and Frontotemporal Degeneration, European Neurology, American Journal of Neurodegenerative Diseases, Frontiers in Neurology, and Exploration of Neuroprotective Therapy. V.S., N.T. and F.V. acknowledge the support of Italian Ministry of Health (Ricerca Corrente/Ricerca Finalizzata; Hub Life Science-Diagnostica Avanzata (HLS-DA), PNC-E3-2022-23683266, the Italian Ministry of Health within the Complementary National Plan Innovative Health Ecosystem) and the Italian Ministry of Education and Research ("Dipartimenti di Eccellenza" Program 2023-2027, Department of Pathophysiology and Transplanta-tion, Università degli Studi di Milano), unrelated to the work presented in this paper. C.S. received support from the Doris Ruess Stiftung, unre-lated to the work presented in this paper. M.O. received research sup-port from the German Federal Ministry of Education and Research (projects: FTLDc 01GI1007A), the EU Moodmarker programme (01EW2008), the ALS Association, the foundation of the state Baden-Württemberg (D.3830), Boehringer Ingelheim Ulm University BioCenter (D.5009), and the Thierry Latran Foundation and EU-MIRIADE and the Roux-programme of the Martin Luther University Halle (Saale); M.O. received consulting fees from Biogen, Axon, Roche, and Grifols; and participates on the Biogen ATLAS trial board, all unrelated to the work presented in this paper; is a speaker of the german FTLD consortium, is involved in an unpaid role with the German Society for CSF Diagnostics and Neurochemistry, and is involved without pay with the Society for CSF Diagnostics and Neurochemistry. M.O., P.O., and S.H. are co-inventors of a patent application for using beta-synuclein measurement in blood. The other authors report no disclosures relevant to the manuscript.
