## [Transparent Peer Review file · Nature Communications]

Phosphorylated tau 181 and 217 are elevated in serum and muscle of patients with amyotrophic lateral sclerosis

Corresponding Author: Professor Markus Otto

Version 0:

Reviewer comments:

Reviewer #1

(Remarks to the Author)

This work by Abu-Rumeileh and colleagues provides compelling evidence on the presence of tauopathy markers in the peripheral tissues of ALS patients, namely the blood serum and the muscle; moreover, it shows similar serum levels of tau species in AD and ALS patients, raising questions on the biological specificity of tau markers for AD, as recent diagnostic criteria instead do.

The study is well conducted, with appropriate methodology and the quality of the report is good.

The main strengths of the work are the sample size, the multicenter design with validation cohorts, and the commendable translational multi-matrices approach, which allows for further insights into the source of blood tau.

In general, there are no significant criticisms and the publication is recommended. However, some points could be enhanced.

In the introduction, the authors should provide a more robust rationale for comparing ALS vs AD blood p-tau species since there are no main differential diagnosis issues between these two conditions, and there is only doubt on the biological specificity of tauopathy markers. As well, some concepts (e.g., T2 core criteria) should be better put in context, as they are not immediately accessible to the audience.

Regarding patient selection, there is no mention of potential comorbidity that might have affected experimental findings and no exclusion criteria.

Regarding muscular biopsy, it could be helpful to specify which muscle the sample was obtained from. Was it a standard site in every patient or the most affected one?

In discussion, authors might consider that pathological tau species have been found in peripheral tissues derived from patients with other tauopathies, namely PSP, and specifically in the skin (<https://doi.org/10.1093/brain/awac161>), the nerves (<https://doi.org/10.1093/brain/awad381>) and the blood (<https://doi.org/10.1002/mds.30009>), supporting the localization of tau out of the CNS, which is the main result of the present work.

The multicenter nature of the study should be highlighted in the abstract.

Reviewer #2

(Remarks to the Author)

This manuscript explores the differences in phosphorylated tau in serum and muscle between ALS, AD, and Control groups. The authors report elevations in serum pT181 and pT217, suggesting that these changes might reflect protein release from denervated muscle fibers and potentially challenge the specificity of these biomarkers for AD pathology. However, several methodological gaps and assumptions limit the clarity of the manuscript, making it difficult to evaluate the manuscript fully. Some revisions are recommended before further consideration.

1. Title- the current title is somehow too broad. Since the manuscript focuses primarily on pTau181 and with limited discussion on the other ptau, the authors might want to refine the title to be more specific, also instead of 'phosphorylated tau isoforms', it might be better to use the word "phosphorylated tau site" or "phosphorylated tau peptides";
2. Control group- the control groups appear to be age-match, and in this manuscript, the authors also try to discuss the age impact. It might be beneficial to include young normal control for further examine the potential impact of age;
3. Figure for CSF NfH and Serum NfL- On page #5, the comparison of CSF NfH and Serum NfL with pT181 and pT217 would be strengthened by adding supplementary figures;
4. ALS Patients with AD co-pathology- On page #5, the authors mentioned ALS with AD co-pathology, it could be better to

highlight these patients in Figure 1A-D, which should be nice;

5. pTau sites in full-length tau- The discussion on page#7 regarding ptau site identified via mass spectrometric analysis of muscle biopsies, how is the accuracy of these measurements? additional data supporting these findings could clarify the discussion. Also, when concluding the increase of pTau site in ALS might need to consider the total tau level, is it related to the increasing total tau and ptau level or just increasing ptau%?

6. Serum pT217 measurement- in the method section, particularly the blood and CSF analysis section, the description of the serum pTau 217 measurement process could be clearer, it is somehow difficult to follow in the current manuscript;

7. control cohort for muscle biopsies-in the immunoreactivity analysis of muscle biopsies, the cohort includes 13ALS and 14 disease controls, which could be ideal if some health control and AD cases could be included in the comparison;

8. Chromatographic Data for ptau isoforms- in Figure 3A-B, it could be helpful to provide chromatograph of pT181 and pT217. Besides, it would be nice if similar information could be provided for the other specific ptau site generated from full-length tau to support the authors' point;

9. Tau isoform comparisons-in Figure 3C, it could be helpful to present a side-by-side comparison of 2N4R small tau to the full-length tau isoform, by highlighting the specific regions, such as 1N, 2N, and MTBR to aid in visualizing the structural differences

Reviewer #3

(Remarks to the Author)

Authors described the p-tau condition of 90 ALS patients. They analyze p-tau 181 & 217 with multicenter using serum and muscle tissue. They also analyze serum NfL and CSF NfH. P-tau was detected in atrophic muscle fibres.

Major points

1. Although this is a multicenter study, sample size is not so large compared to previous reports.

2. Although NfH & L are surveyed, authors don't describe about this in Discussion. Authors should discuss the meaning of NfH & L in ALS. Do authors recommend not to survey NfH & L in ALS?

3. I think reliance of muscle staining on a one p-tau antibody is not sufficient. Using another p-tau antibody, authors should confirm the reliance (at supplemental section).

Minor point

I can't understand why % of normal CK case is only 29.6%. Median (IQR) is 69.0 (62.7-147.0), and normal values <171 U/L. If this is true, I think at least 75% cases are within normal.

Version 1:

Reviewer comments:

Reviewer #1

(Remarks to the Author)

The authors addressed all the points , congrats for the work

Reviewer #2

(Remarks to the Author)

I appreciate the authors' efforts in addressing the comments and suggestions provided during the initial review. The revisions demonstrate a thoughtful and comprehensive response to the feedback, particularly with the inclusion of additional control groups and other key updates, which have strengthened the overall quality and clarity of the manuscript.

At this stage, I have no major concerns regarding the manuscript. The authors have addressed the critical points raised during the review process, and the revised manuscript could be considered for acceptance.

However, I have one minor suggestion for further improvement: the use of the term "peptides" in reference to p-tau throughout the manuscript may not always be necessary. For instance, "p-Tau217" and "p-Tau 181" are sufficiently specific and clear without including the word "peptides." Similarly, using the term "p-tau species" could provide flexibility and consistency in terminology. Adopting this approach may enhance readability and align better with standard nomenclature.

Reviewer #3

(Remarks to the Author)

The authors appropriately addressed comments.

Reviewer's Comments:

Reviewer #1 (Remarks to the Author)

This work by Abu-Rumeileh and colleagues provides compelling evidence on the presence of tauopathy markers in the peripheral tissues of ALS patients, namely the blood serum and the muscle; moreover, it shows similar serum levels of tau species in AD and ALS patients, raising questions on the biological specificity of tau markers for AD, as recent diagnostic criteria instead do.

The study is well conducted, with appropriate methodology and the quality of the report is good. The main strengths of the work are the sample size, the multicenter design with validation cohorts, and the commendable translational multi-matrices approach, which allows for further insights into the source of blood tau.

In general, there are no significant criticisms and the publication is recommended. However, some points could be enhanced.

We are very grateful to the reviewer for his positive comments.

In the introduction, the authors should provide a more robust rationale for comparing ALS vs AD blood p-tau species since there are no main differential diagnosis issues between these two conditions, and there is only doubt on the biological specificity of tauopathy markers.

We thank the reviewer for the comment. We have changed the introduction according to the reviewer's suggestions (pages 3-4).

As well, some concepts (e.g., T2 core criteria) should be better put in context, as they are not immediately accessible to the audience.

We have amended the introduction in line with the reviewer's suggestions adding more details (page 3). P-tau 217 and p-tau 181 are biomarkers of initial AD neuropathological changes (Core 1 T1 category according to the Revised criteria for diagnosis and staging of AD) (Jack et al. 2024). Indeed, the two peptide levels become abnormal early, around the same time as amyloid positron emission tomography (PET) and before tau PET, and possibly reflect the secretion of phosphorylated mid-region tau fragments in response to amyloid plaques or to soluble A β species, thus linking A β proteinopathy to early tau proteinopathy.

We would like to apologise for mistakenly writing "T2 biomarkers" instead of "T1".

Regarding patient selection, there is no mention of potential comorbidity that might have affected experimental findings and no exclusion criteria.

The reviewer's observations are correct. We highlight this issue as a limitation in the discussion (page 10).

Regarding muscular biopsy, it could be helpful to specify which muscle the sample was obtained from. Was it a standard site in every patient or the most affected one?

There was no standard location for muscle biopsies. Rather, the choice of muscle was made on a subject by subject basis, as dictated by the clinical affection. Information on the specific muscle from which the sample was taken is provided in the supplementary table 8.

In discussion, authors might consider that pathological tau species have been found in peripheral tissues derived from patients with other tauopathies, namely PSP, and specifically in the skin (<https://doi.org/10.1093/brain/awac161>) , the nerves (<https://doi.org/10.1093/brain/awad381>) and

the blood (<https://doi.org/10.1002/mds.30009>), supporting the localization of tau out of the CNS, which is the main result of the present work.

We thank the reviewer for the suggestion and modify the discussion accordingly (page 10).

The multicenter nature of the study should be highlighted in the abstract.

We thank the reviewer for the suggestion and modify the abstract accordingly (page 2).

Reviewer #2 (Remarks to the Author)

This manuscript explores the differences in phosphorylated tau in serum and muscle between ALS, AD, and Control groups. The authors report elevations in serum pT181 and pT217, suggesting that these changes might reflect protein release from denervated muscle fibers and potentially challenge the specificity of these biomarkers for AD pathology. However, several methodological gaps and assumptions limit the clarity of the manuscript, making it difficult to evaluate the manuscript fully. Some revisions are recommended before further consideration.

1. Title- the current title is somehow too broad. Since the manuscript focuses primarily on pTau181 and with limited discussion on the other ptau, the authors might want to refine the title to be more specific, also instead of 'phosphorylated tau isoforms', it might be better to use the word "phosphorylated tau site" or "phosphorylated tau peptides";

We modified the title accordingly and adopted the term "peptides". We have also changed the term throughout the text.

2. Control group- the control groups appear to be age-match, and in this manuscript, the authors also try to discuss the age impact. It might be beneficial to include young normal control for further examine the potential impact of age;

As suggested by the reviewer, we included serum samples from 23 young healthy controls (HC) in the study, aged less than 50 years (mean age 35.7 ± 7.7 years, min-max 22-48 years), with normal Mini-Mental State Examination score and no history of neurological disease. Due to ethical concerns, lumbar puncture with CSF AD core biomarker assessment was not performed in these individuals to exclude incidental AD co-pathology. Nevertheless, all these subjects had serum NfL within the normal range according to our in-house cut-off values. Furthermore, given their age < 50, the occurrence of incidental AD co-pathology in this group should be considered unlikely (Braak et al. 2011). We added this information in the method section (page 12).

Accordingly, we have renamed the group of non-neurodegenerative disease controls as disease controls throughout the text.

In the healthy control group, in addition to NfL, we also measured serum p-tau 181, p-tau 217 and total tau (see supplementary Table 1 for median values). We then examined the associations between age and blood p-tau tau peptides and t-tau in the disease control (n=99) and healthy control (n=23) groups and in the combined control group (n=122). The effect of age on serum NfL was not investigated, since it has been extensively explored in the literature and was beyond the scope of this study.

All the new analyses have been reported in the supplementary results. Interestingly, we found moderate to strong associations between age and p-tau 181 (Spearman's $r=0.556$, $p<0.001$) or p-tau 217 ($r=0.702$, $p<0.001$) in the combined control group (see supplementary results, page 1). We also add a Supplementary Figure 1 showing these associations.

In addition, we have included biomarker group comparisons including healthy controls in the supplementary results (pages 1-2). Given the impact of age on biomarker levels, our analyses were already age-adjusted in the first version of the manuscript.

In the Discussion, we briefly expanded on the discussion of the potential effect of age on p-tau peptides in serum and muscle biopsies (page 9). Here, the finding of increasing nuclear p-tau 181 and p-tau 217 reactivity with age in controls is consistent with the strong correlations between age and serum p-tau peptides in our control group and may suggest an age- or senescence-related effect on the release of p-tau peptides from muscle tissue, which may also be independent of disease states and warrants further investigation.

3. Figure for CSF NfH and Serum NfL- On page #5, the comparison of CSF NfH and Serum NfL with pT181 and pT217 would be strengthened by adding supplementary figures;

A supplementary figure (now Supplementary Figure 2) comparing serum markers, namely NfL, p-tau 181 and p-tau 217 levels across diagnostic groups was already included in the first version of the manuscript. We improve this figure by adding also serum t-tau.

For completeness we have also added a new supplementary figure (Supplementary Figure 3), showing the distribution of CSF NfH, serum NfL and serum t-tau levels in the whole cohort and in the individual cohorts.

4. ALS Patients with AD co-pathology- On page #5, the authors mentioned ALS with AD co-pathology, it could be better to highlight these patients in Figure 1A-D, which should be nice;

We have modified Figures 1A and 1C (whole cohort) by showing serum p-tau 181 and p-tau 217 levels in 1) ALS (entire group), 2) ALS A+ (with AD co-pathology), 2) ALS A- (without AD co-pathology); 3) AD and 4) disease controls (DCo). Due to the relatively small number of ALS cases with AD co-pathology in each cohort, we decided not to show them in Figures 1B and 1D.

The same subdivision of ALS cases has been adopted in the supplementary Figure 3 on serum t-tau, serum NfL and CSF NfH distribution.

5. pTau sites in full-length tau- The discussion on page#7 regarding ptau site identified via mass spectrometric analysis of muscle biopsies, how is the accuracy of these measurements? additional data supporting these findings could clarify the discussion.

To support the accuracy of our results, we have included the respective quality values of the measurements for all phosphorylated tau sites in the new Supplementary Table 7. Furthermore, we added further information in the method (page 15) and results sections (page 7).

In detail, stringent filtering was applied to the phosphosites identified by our mass spectrometry approach. All sites had to pass filtering for a false discovery rate of less than 0.01. The identified phosphosites were filtered based on the phosphorylated protein. Only sites identified in the protein sequence of any tau isoform were further analysed. The site localization probability had to be > 0.95 for singly phosphorylated peptides or between 0.475 and 0.525 for adjacent acceptor sites in doubly phosphorylated peptides. The overall posterior error probability had to be less than 0.02. In a final check, the identification score (delta score relative to the unmodified peptide) had to be at least 40 points greater than for the unphosphorylated counterpart.

Nevertheless, as the main text should not exceed 5000 words, we have moved a part of MS methods into supplementary methods.

Also, when concluding the increase of pTau site in ALS might need to consider the total tau level, is it related to the increasing total tau and ptau level or just increasing ptau%?

To test whether the increase in p-tau peptides in the blood of ALS patients could be related to the increase in total tau or to the relative increase in p-tau, we measured total tau in muscle biopsies by mass spectrometry (MS) and in serum samples with the Simoa platform.

The stochastic nature of data-dependent acquisition combined with our small sample size (5 ALS and 5 disease controls) allowed an MS quantification of the total protein (total tau abundance) in muscle biopsies. In contrast, our MS-based phosphoproteomics provided only qualitative information on p-tau 181 and p-tau 217 and did not allow MS-based quantification of individual p-tau sites. In the result section we have added quantitative data obtained by mass spectrometry analysis of total tau abundance in muscle biopsies (page 7). Here we found no significant difference in total tau abundance between ALS patients and controls in muscle biopsies. We have also added a Supplementary Figure 5 regarding this comparison.

We also measured total tau in serum using the Simoa platform in the vast majority of our cohort (N=357 out of 385, including old and new cases, see response to Reviewer 3). We have included these new data in the manuscript (page 5, Table 1) and in the supplementary materials (supplementary results page 2, supplementary tables 1-6, supplementary figures 2 and 3). In this regard, we found similar serum total tau levels in ALS and disease controls, consistent with previous literature on blood t-tau in ALS (Kasai et al. 2019; Agah et al. 2024). Notably, we found relatively similar or lower levels of serum t-tau in relation to the corresponding p-tau peptide levels. However, these data are consistent with previous literature (Ni et al. 2023, Bentivenga et al. 2024) and may be due to the different sensitivities of the Simoa assays.

Taken together, our muscle and serum data suggest that the increase in blood p-tau peptides in ALS is probably related to the relative increase in p-tau and not to a general increase in total tau levels, as total tau concentrations in muscle tissue and blood did not differ between ALS and disease controls. We added this topic to the discussion (pages 8-9).

6. Serum pT217 measurement- in the method section, particularly the blood and CSF analysis section, the description of the serum pTau 217 measurement process could be clearer, it is somehow difficult to follow in the current manuscript;

We apologise for the lack of clearness, and we modified the method section trying to be more clearer concerning the measurements of p-tau 217. However, as the main text should not exceed 5000 words, we have moved all this topic into supplementary methods.

7. control cohort for muscle biopsies- in the immunoreactivity analysis of muscle biopsies, the cohort includes 13ALS and 14 disease controls, which could be ideal if some health control and AD cases could be included in the comparison;

Muscle biopsies are not routinely performed in subjects with Alzheimer's disease or in healthy controls because of ethical concerns about the use of such invasive procedures. Therefore, this information cannot be provided.

As described in Methods and Supplementary Table 8, subjects who underwent muscle biopsy as part of a diagnostic work-up for myalgia, cramps and exercise intolerance were referred to as 'disease controls'. They underwent a complete clinical examination, electromyography and myohistological investigations all of which showed no pathological findings. We believe that these are the best controls we can currently obtain in accordance with our local ethical guidelines. We improved the respective description in the method section (pages 13-14).

The lack of muscle biopsies from AD cases and healthy controls has been discussed as a limitation of the study (page 11).

8. Chromatographic Data for ptau isoforms- in Figure 3A-B, it could be helpful to provide chromatograph of pT181 and pT217.

We added chromatographic data for p-tau peptides in Figure 3 A-B.

Besides, it would be nice if similar information could be provided for the other specific ptau site generated from full-length tau to support the authors' point;

Spectra for the tau phosphosites reliably identified in our analyses but not individually mentioned in the main text or shown in Figure 3 are provided in the newly added Supplementary Figure 4.

9. Tau isoform comparisons-in Figure 3C, it could be helpful to present a side-by-side comparison of 2N4R small tau to the full-length tau isoform, by highlighting the specific regions, such as 1N, 2N, and MTBR to aid in visualizing the structural differences

We have adapted Figure 3C to show the two tau isoforms side-by-side and have added the protein domains and the localisation of the N-terminal insert and repeats in the microtubule-binding region.

Reviewer #3 (Remarks to the Author):

Authors described the p-tau condition of 90 ALS patients. They analyze p-tau 181 & 217 with multicenter using serum and muscle tissue. They also analyze serum NfL and CSF NfH. P-tau was detected in atrophic muscle fibres.

Major points

1. Although this is a multicenter study, sample size is not so large compared to previous reports.

We agree with the reviewer's observation and have therefore included additional cases of ALS (n=62), AD (n=44) and disease controls (n=20) in the current version of the manuscript. The final cohort now includes a total of 152 ALS, 111 AD and 99 disease controls.

These numbers are now larger than in previous publications (Cousins et al. 2022: 130 ALS, 79 AD and 26 controls; Vacchiano et al. 2023: 148 ALS, 88 AD and 60 controls).

In addition, as suggested by Reviewer 1, we also included a group of 23 young healthy controls (<50 years) (see response to Reviewer 1).

We measured all the previous biomarkers and a new one (serum total tau, see answers to Reviewer 2) and repeated all the analyses in this larger cohort. We hope that these numbers meet the reviewer's and editor's request.

2. Although NfH & L are surveyed, authors don't describe about this in Discussion. Authors should discuss the meaning of NfH & L in ALS. Do authors recommend not to survey NfH & L in ALS?

We apologise for not describing the role of NfH and NfL in ALS, but this was only due to a word count issue. Our research group has performed many studies (e.g.: *Steinacker et al. 2017*, *Verde et al. 2019*, *Abu Rumeileh et al. 2020*, *Halbgebauer et al. 2022*) on the diagnostic and prognostic role of NfL and NfH in ALS and we strongly believe in the high performance of these biomarkers for the multimodal assessment of ALS. In the new version of the manuscript, we have included a paragraph on neurofilaments in ALS in the discussion of the manuscript (page 10).

3. I think reliance of muscle staining on a one p-tau antibody is not sufficient. Using another p-tau antibody, authors should confirm the reliance (at supplemental section).

For muscle staining of p-tau 181 and p-tau 217 we used antibodies (p-tau 181, Thermo Fisher Scientific, MA, USA; AT270 #MN1050; 1:50; p-tau 217, Thermo Fisher Scientific, MA, USA; #44-744; 1:50) that have been previously used in the vast majority of other publications on IHC; see for example: Kim et al. *Acta neuropathologica* 2023; Saunders et al. *Brain communications* 2023; Ercan

et al. Molecular neurodegeneration 2017; Saroja et al. Alzheimers Dementia 2022. For other citations please refer to: <https://www.thermofisher.com/antibody/product/Phospho-Tau-Thr181-Antibody-clone-AT270-Monoclonal/MN1050>; <https://www.thermofisher.com/antibody/product/Phospho-Tau-Thr217-Antibody-Polyclonal/44-744?imageId=59100>.

At the time of performing the muscle biopsy analyses, we had already searched for alternative antibodies directed against p-tau 181 and p-tau 217 to test the reliability of the staining, but, unfortunately, very few had been previously studied in the literature. Moreover, all of them have been studied in a very limited number of publications on IHC (lower number of citations than for the two antibodies reported above).

Therefore, in order to exclude the possibility of non-specific binding of the p-tau antibodies used, we decided to also study the presence of p-tau peptides in muscle biopsies by mass spectrometry, which is a highly specific and reliable method. Here we confirmed the presence of p-tau 181 and p-tau 217 peptides in ALS and control muscle tissue.

To support the accuracy of our mass spectrometry results, we have included the respective quality values of the measurements for all phosphorylated tau sites in the new supplementary table 7 (for more details see responses to reviewer 2).

Taking all these data together, we strongly believe that the evidence in the literature, together with our concordant mass spectrometry data, already supports the reliability of the p-tau antibodies we used.

Minor point

I can't understand why % of normal CK case is only 29.6%. Median (IQR) is 69.0 (62.7-147.0), and normal values <171 U/L. If this is true, I think at least 75% cases are within normal.

The reviewer is right. We made a mistake and at least 75% of the cases were in the normal range. However, we have included new cases in the new manuscript version and changed all the table values accordingly. The median (IQR) for CK was 133.0 (65.9-253.4) U/L with 59.3% of the cases being in the normal range.

In the larger ALS cohort from Halle and Milan, we were not able to confirm a significant association between myoglobin and p-tau peptides; however we found a significant association between the latter and troponin T, another marker of striated muscle damage.

Reviewer's Comments:

Reviewer #1 (Remarks to the Author):

The authors addressed all the points , congrats for the work

We thank the reviewer for the positive comments.

Reviewer #2 (Remarks to the Author):

I appreciate the authors' efforts in addressing the comments and suggestions provided during the initial review. The revisions demonstrate a thoughtful and comprehensive response to the feedback, particularly with the inclusion of additional control groups and other key updates, which have strengthened the overall quality and clarity of the manuscript. At this stage, I have no major concerns regarding the manuscript. The authors have addressed the critical points raised during the review process, and the revised manuscript could be considered for acceptance.

We thank the reviewer for the positive comments.

However, I have one minor suggestion for further improvement: the use of the term "peptides" in reference to p-tau throughout the manuscript may not always be necessary. For instance, "p-Tau217" and "p-Tau 181" are sufficiently specific and clear without including the word "peptides." Similarly, using the term "p-tau species" could provide flexibility and consistency in terminology. Adopting this approach may enhance readability and align better with standard nomenclature.

We modified the words throughout the manuscript according to the reviewer suggestions.

Reviewer #3 (Remarks to the Author):

The authors appropriately addressed comments.

We thank the reviewer for the positive comments.